# Attribution of credit in acknowledgements: The case of systematic reviews in medicine

Rossella Salandra[1]*, Marisa Miraldo[2], Paola Criscuolo[3]

**1** Strategy & Organisation Division, School of Management, University of Bath, Bath, United Kingdom, **2** Department of Economics & Public Policy and Centre for Health Economics & Policy Innovation, Imperial College Business School, London, United Kingdom, **3** Department of Management & Entrepreneurship, Imperial College Business School, London, United Kingdom

* r.salandra@bath.ac.uk

## Abstract

Scientific research increasingly benefits from the work of non-author contributors, who engage in valuable activities but do not meet authorship criteria. The support of these contributors remains invisible unless it is recorded in acknowledgement texts, which has implications for research transparency and fair attribution of credit. While much research has examined authorship, acknowledgements have been largely neglected. We explore whether acknowledgements omit deserving contributors and the heterogeneity of omissions by gender and race. We focus on the medical field, which is often a forerunner in terms of attribution of credit, and consider acknowledgements in Cochrane systematic reviews. These reviews are governed by unique guidance that allows us to determine whether a contributor receives due credit. We find that as many as 40% of the eligible reviews in our sample (those that should have acknowledged prior authors) did not appropriately acknowledge non-author contributors. Non-White contributors were more likely to be missing from the acknowledgements. This disparity cannot be explained by non-White contributors being more likely to perform minor/technical review tasks or being in predominantly White research teams or teams led by White scientists. Instead, the effect is driven by geographical disparities, with non-White contributors being more likely to be deprived of due acknowledgements in reviews from Asia, South America and Africa. Furthermore, in most cases, all contributors were omitted from the acknowledgements, rather than specific contributors being excluded alone. Taken together, we provide novel evidence of some racial disparities in credit attribution in acknowledgements. Reassuringly, these appear to be driven by poor acknowledgement practice and geographical disparities rather than targeted exclusion.

## Introduction

Acknowledgements are a form of academic recognition included in the 'reward triangle' alongside authorship and citations [1]. Among the elements of this triangle,

**Data availability statement:** The data underlying the results presented in the study are available from the University of Bath Research Data Archive: https://doi.org/10.15125/BATH-01609.

**Funding:** The author(s) received no specific funding for this work.

**Competing interests:** The authors have declared that no competing interests exist.

authorship has historically been regarded as the primary mechanism for accumulating academic prestige, and it has been at the basis of the scientific reward system [2,3]. However, trends toward larger scientific teams and the bureaucratisation of science [4,5] have challenged the established notion of authorship, leading to the increasing recognition that all contributors, even beyond the authorship team, should be acknowledged. Acknowledgements offer a way to express gratitude to individuals who didn't make contributions that rise to the level of authorship, but supported the work with conceptual, editorial, financial, moral or instrumental/technical support [6]. They differ from authorship and citations due to their more subtle and personal nature [7,8]. Moreover, acknowledgements are not mandatory and are generally left to the discretion of the authors, hence reflecting voluntary acts of recognition. As with authorship, the decision to mention someone in the acknowledgements can be motivated by many reasons, ranging from sincere gratitude to strategic 'name-dropping'.

In this paper, we set to investigate whether non-author contributors are properly credited in acknowledgement texts. This warrants attention, given the increasing importance of non-author contributors for conducting research. Beyond being a 'scholar's courtesy' [6], proper attribution in the acknowledgements is imperative to attest to the fairness of the research process. A mention in the acknowledgement section "*is not a worthless act; on the contrary, it is a reflection of the cooperative context in which research takes place*" ( [9]: p. 2). Moreover, the creation of acknowledgement databases has increased interest towards acknowledgement mentions as a metric to measure researchers' contributions [7,8,10]. We are particularly interested in the acknowledgement of scholars who are minorities in terms of gender and race, for whom a lack of attribution might intensify disparities occurring in other parts of the scientific system, e.g., authorship [11].

It is essential to note that throughout the paper, we refer to gender rather than sex, as our analysis primarily uses a gender detection tool, Gender-API, which infers gender from names. This is consistently described as one of the most accurate gender-detection tools, with an estimated accuracy of 96% [12,13]. However, it also has weaknesses, notably the assignment of gender on a binary scale. Moreover, as our analysis uses full names to infer race, leveraging the prediction package Ethnicolr, and we don't have richer data (e.g., self-identified data on ethnicity or nationality), we refer to race. Ethnicolr is commonly used but has an imperfect accuracy of 83% (https://ethnicolr.readthedocs.io/ethnicolr.html#evaluation).

Assessing whether acknowledgements mention or omit deserving contributors is challenging because they are uncodified texts, with significant variations among journals or disciplines [14,15]. More broadly, a key empirical challenge in studying the attribution of credit arises from the fact that who is or is not deserving of recognition is often difficult to identify, as effort is not directly observable. To address these issues, we focus on systematic reviews of healthcare interventions published by Cochrane, an international not-for-profit research organisation. Cochrane reviews are subject to regular updates to incorporate the latest medical evidence, so several versions of the same review can exist. According to Cochrane guidance, all authors of the preceding version of a review are considered non-author contributors and need to

be acknowledged [16]. This setting, therefore, allows us to observe whether non-author contributors receive credit in the acknowledgements of the updated review.

We use data from 2,091 Cochrane reviews, subject to at least one update between 2003 and 2019. For each review-review pair, where each pair contains the original and updated versions of a given review, we observe whether non-author contributors are mentioned in the acknowledgements of the updated review. We find that 40% of the eligible reviews in our sample (those that should have acknowledged prior authors) did not fully acknowledge non-author contributors, even despite the guidance. We focus on salient characteristics typically shaping unwarranted variation in the attribution of credit in science, particularly gender and race [17–19]. Gender does not appear to affect omission from acknowledgements. However, we document racial disparities, with non-White contributors in our sample being less likely to be acknowledged than their White counterparts. Being more likely to contribute to minor/technical review tasks, being on a White-dominated research team or in a team led by White scientists did not increase the odds of being omitted from the acknowledgements for the non-White group. Instead, we found that non-White contributors were less likely to be missing from the acknowledgements in White-dominated teams but more likely to be missing from acknowledgements when the corresponding author was non-White. We also observe geographical disparities, with non-White authors in reviews from Asia, South America and Africa being more likely to be neglected in the acknowledgements. Reassuringly, most of the 'forgotten' contributors (88%) were omitted in the acknowledgements in reviews that failed to mention *all* contributors and only 12% were omitted in reviews in which someone else was acknowledged.

In summary, in answering our research question "*Do acknowledgements omit deserving contributors and are potential omissions shaped by the gender and race of the focal contributor?*", our optimistic conclusion is that the disparities we observe are likely to reflect poor acknowledgement practices or a lack of awareness of good acknowledgement guidance rather than targeted omissions of certain groups of contributors. We discuss the implications of our findings for research on attribution of credit and inequalities in science. The study's main contribution is to shed light on the recognition of non-author collaborators and to explore whether disparities observed elsewhere, such as in authorship and citations, extend to acknowledgement texts. In particular, by leveraging updated Cochrane reviews, in which acknowledgement guidance is explicit, we provide rare evidence on whether deserving non-author contributors are recognised and whether omissions vary by gender and race. Situating acknowledgements within the scientific reward system, we show that disparities can persist even in highly codified contexts, underscoring the need for enforceable and straightforward guidance on acknowledgements. From a science policy perspective, we hope this study contributes to a better understanding of how editorial guidance on acknowledgements may amplify or counteract biases in attribution of credit.

## Related work

### Attribution of credit for non-author contributors via acknowledgements

Authorship has traditionally been central to the reward system in science [2,3], and accordingly, objective criteria for authorship attribution are often in place. In the medical field, for example, the International Committee of Medical Journal Editors (ICMJE) recommends that all named authors must have made substantial contributions to the work's conception, participated in drafting or revising it, and agreed to be accountable for its content. The established notion of authorship, however, is increasingly being challenged, leading many publishers to adopt contributorship statements, such as CRediT [20], and to encourage appropriate attribution of non-author contributors. These contributors are typically acknowledged in the publication's acknowledgement section.

Unlike authorship and citations, which are institutionalised mechanisms for attributing credit, acknowledgements are voluntary acts of recognition, discretionary, and governed by professional and normative behaviour norms that are largely uncodified and implicit. They can be used to express appreciation for a range of specific individuals, such as reviewers [21], supervisors [22], or family members, as well as institutions, e.g., funding bodies [23,24]. Whereas authorship captures formal collaboration, acknowledgements reveal more informal influences, including moral or personal ones [14]. The

nuanced nature of acknowledgement texts offers a unique window into research practices and the interactions that might not be evident by simply looking at authorship bylines. The analysis of acknowledgement patterns in genetics research, for example, provides helpful insights into the exchange of information and materials that occurs during the research process [25].

As is the case with authorship (see [26] for a systematic review), acknowledgement practices are subject to substantial interdisciplinary differences. The prevalence of acknowledgements, as well as the types of support acknowledged, varies across disciplines [14,15]. For example, most articles in the hard sciences include acknowledgements [25], reflecting the working practices within these fields, e.g., more team-based research approaches. The properties of acknowledgements also vary by discipline, with more elaborate writing in the humanities and social sciences [27]. Differences have also been observed in subject area and journal prestige [28], with the geographical context of publication also influencing the frequency and length of acknowledgements, which are more common and longer in UK and US journals [29].

The decision to acknowledge someone can be motivated by many reasons, leaving leeway for flattery and self-promotion, beyond personal gratitude. Acknowledgements can be used deliberately, and the information they include can be interpreted in terms of both normative and strategic accounts [30]. For example, they might be used to acquire recognition by association or even to avoid certain peer reviewers ("Cite your friend, acknowledge your foe"). Analysis of acknowledgements in student dissertations also suggests that acknowledgements are shaped by strategic choices, e.g., signalling status by associating the author with influential figures [22].

## The under-recognised role of acknowledgements

Acknowledgements have received comparatively less attention relative to other forms of academic reward; for recent reviews, see [8] and [31], leading them to be described as a disregarded "Cinderella" genre [22]. This has been motivated by two main reasons. First, the omission of non-author contributors has traditionally been considered a marginal issue, possibly because a mention in the acknowledgements is not a currency valued as authorship. There is also the expectation that non-author contributors may have different norms, e.g., they may be only marginally interested in being mentioned in a publication [32]. However, the incomplete attribution of credit in the acknowledgements can hamper the transparency of the research process and the allocation of practical and reputational resources of the focal contributors, e.g., assessments of their helpfulness [33]. A mention in the acknowledgement is also a reward and "*intrinsically valuable simply for the pleasure of being acknowledged*" ( [34]: p. 1). In some fields, such as medicine, acknowledgements are also seen as a stepping stone towards authorship, so unjustified omissions can have potential career and well-being implications. Acknowledgements might also signal informal collaboration and belonging to the 'right circles', particularly in specific domains. In the medical field, for example, acknowledgements are included in potential applicants' CVs and are seen as a marker of a promising trajectory. Indeed, acknowledgements are linked to other forms of reward and academic performance [35,36].

Second, acknowledgement texts are often messy and unstandardised, and, as anticipated above, acknowledgement practices can vary across geographies and time [e.g., 29]. Some research areas, such as medicine, which is often at the forefront of attribution of credit initiatives, have more explicit guidance. ICMJE recommendations, for example, state that contributors who do not meet all the authorship criteria should not be listed as authors but should still be acknowledged. Assessing whether the acknowledgements section of a paper is comprehensive and a fair reflection of the support received by non-author contributors is challenging, given the significant variations in guidelines [32]. More broadly, determining who deserves credit in a science project is complex because actual contributions are not always easy to observe. This is likely to be more difficult when the contribution, despite being valuable, is not significant enough to warrant authorship. Furthermore, in the past, acknowledgements were not indexed in bibliometric databases. This has now changed, given the availability of acknowledgement databases [e.g., 24,37].

 

We argue that examining acknowledgements deserves attention, given the growing involvement of non-author collaborators and the pivotal role acknowledgements play in credit attribution. By ensuring that all contributions are correctly recognised, proper attribution through acknowledgements helps maintain the integrity of the research system, promoting transparency and fairness within the research community. Importantly, scientists express unhappiness when not thanked, reflecting an expectation of reciprocity [25]. In this paper, we focus on exploring visibility and acknowledgement for scholars from minority backgrounds, particularly in terms of gender and race.

### The impact of gender and race on acknowledgement practices

Inequalities in authorship and citations have been extensively reviewed in the attribution of credit literature. Concerning gender, there is ample evidence that female scientists are underrepresented in the esteemed first and last author positions, as well as among authors of single-authored papers [38]. While this gap is well established [19], only recent work has determined that women are less likely to be named as authors because their contributions are undervalued [39]. The lack of recognition for female scientists is also visible through the under-citation of their articles [40]. Most studies investigating gender and acknowledgements have focused on the percentage of females being acknowledged, suggesting that gender differences found in authorship might extend to non-authors. Cronin et al. [41] found that while most authors in women's studies are women (93%), only 66% of the acknowledgees are women. In financial economics, female scientists appear less often in the acknowledgements than their male colleagues [42]. Investigating the acknowledgements of articles indexed in the Web of Science, Paul-Hus et al. [43] also found that the percentage of female acknowledgees is lower than that of their male counterparts.

With regard to race, past work suggests that racial and ethnic minority scholars are also unequally rewarded for their contributions. For example, Bertolero et al. [19] found that reference lists tend to include more papers with a White individual in the first and last author position than would be expected. Papers authored by non-White scholars receive fewer citations, with the gap being particularly pronounced for Black authors [44]. Race disparities in acknowledgements have received less attention. Prior work on this topic has also primarily focused on the percentage of minority contributors being acknowledged. For example, examining footnotes in law review articles, Nunna et al. [45] found that White scholars are acknowledged more frequently.

## Methods

This project obtained a favourable ethics opinion from the University of Bath Data & Digital Science Research Ethics Committee (Ref: 1543−1206). Informed consent was not obtained as the project used secondary data that were available in the public domain.

### Cochrane systematic reviews

Our analysis draws on systematic reviews compiled by Cochrane (https://www.cochrane.org), a world-leading not-for-profit organisation that provides systematic reviews in healthcare. Systematic reviews are commonly recognised as one of the foundations of evidence-based clinical decision making [46] and shape the development of health policies and clinical guidelines. Cochrane is the leading institution in the field, with representation across more than 190 countries. Cochrane reviews combine the results of several primary research studies, typically clinical trials, applying reproducible approaches to generate reliable estimates of the effectiveness of interventions. They are peer-reviewed and require considerable resources, such as selecting relevant studies, critically appraising the evidence, and conducting meta-analyses. The Cochrane Handbook is the official guide that Cochrane authors consult for guidance on methods. Data from Cochrane have been used in other papers, for example, to examine the antecedents of selective reporting [47].

Cochrane reviews have two interesting features for studying credit attribution: i. they are routinely updated, and ii. according to Cochrane editorial policies, individuals who helped in the first review as authors should be acknowledged in the updated review.

**Updating reviews.**  Cochrane dedicates substantial resources to ensure that all reviews are up to date, a necessity as medical evidence continually evolves. An updated review is a "*new edition of a published systematic review with changes that can include new data, new methods, or new analyses to the previous edition*" ( [48]: p.2). The availability of an original and an updated review allows us to observe how author and non-author contributorship are handled when a review is revised over time. For example, consider the Cochrane review titled "Biopsy versus resection for high-grade glioma" (Fig A in S1 File). The review was published in 2000 and underwent an update in 2019. The 'Version history' information for this review shows that some of the authors in the 2000 review are not listed as authors in the 2019 review. These changes can be expected due to the natural processes of reconfiguration of the review team.

**Guidance for attribution of credit in updated reviews.**  According to Cochrane guidance on credit attribution, if an author is no longer involved in a review once it is updated, they should not be listed as an author of the updated review. However, they should be named in the acknowledgements section of the updated review. In particular, Section IV.5.1 of the Cochrane Handbook states that "*If an author is no longer actively contributing to... an updated review, the author should not be listed in the by-line of the new version and should be named in the Acknowledgements section*". This is deemed fair because updated reviews typically rely on content created by the original review team — e.g., they may use search and critical appraisal methodologies and expand upon the critical appraisal carried out during the first review. The individuals in the original team are considered non-author contributors and deserve to be acknowledged in line with ICMJE regulations. Thanks to this unique guidance, we can determine whether someone who has made a non-author contribution to the review project — i.e., someone listed as an author in the original review — did or did not receive credit for it.

Although the guidance is currently in place, we do not know precisely when it was first introduced. We found a reference to the guidance in a 2014 PLoS Medicine paper that describes the norms for attribution of contribution in 'living' reviews [49]. The work of the panel for updating guidance for systematic reviews (PUGs) group led to the development of guidance and a checklist discussed in a seminal article published in the British Medical Journal in 2016 [48]. Tracking when Cochrane explicitly adopted the guidance is more complicated. As all authors of the 2014 paper are contributors of the Cochrane Collaboration, and equally, most PUGs participants declared to be affiliated with Cochrane, we assumed that the guidance has been well-known within the Cochrane network at least since 2014.

## Sample construction

We extracted key information for all the reviews published in the Cochrane Database of Systematic Reviews (CDSR) and that had been subject to an update as of June 2019, corresponding to 2,127 unique review-review pairs. We removed from the sample 36 reviews in which the acknowledgements section was missing. This left us with 2,091 review pairs corresponding to 8,267 non-unique authors — the authors are non-unique because the same author can appear in more than one review. These authors were searched on Elsevier's Scopus to retrieve their publication portfolios and citations. To retrieve bibliographic data from Scopus, we used the Scopus Application Programming Interface (API). One of the advantages of deriving our data from Scopus is its disambiguation of authors along with the assignment of unique Author IDs. This ensures that authors with identical names are treated as distinct individuals and prevents the merging of records corresponding to different contributors. In the main analysis, we used a final sample of 7,752 authors for whom race, gender and publication information were available. A flow chart for the sample construction is summarised in Fig B in S1 File.

  

## Omission from authorship and acknowledgements

According to Cochrane's attribution of credit guidance, each author of the original review should be acknowledged in the updated version. However, an acknowledgement is unnecessary when an author remains on the authorship list of the updated review. Our study is, therefore, designed as a funnel of two sequential stages where an individual must pass through a first stage (being omitted in the authorship list) to reach the second stage (being omitted in the acknowledgements). In the first stage, we determine whether the author of a review is (or is not) included in the authorship list of the updated review version. Specifically, Omission from authorship takes the value of 1 if the focal author is absent from the authorship list of the updated review and 0 otherwise. In the second stage, we capture whether the author of a given review is absent from the acknowledgement in the updated version of the review, conditional on being absent from authorship. Omission from acknowledgements takes the value of 1 if the focal author is not mentioned in the acknowledgements of the updated review and 0 otherwise.

To build this variable, we searched for the author's surname and name in the acknowledgements. We started by using the surname alone as some acknowledgements only refer to the first name and initial, e.g., "We thank J. Smith". We manually checked all acknowledgements to identify any cases in which surnames had been misspelt and to remove any acknowledgees with the same surname but a different first name. We then searched for the author's name and evaluated, case by case, whether an individual was listed in the acknowledgements with a different surname (e.g., women changing their maiden name after getting married). Finally, we determined whether the acknowledgements contained sentences such as "We acknowledge the work of the author team for the first version of the review". Even if the individuals were not named, we treat these cases (n = 25 out of 7,752 observations, or 0.32% of the sample) as a proper acknowledgement (Omission from acknowledgements = 0).

## Methodological approach

Our methodological approach is summarised in Fig C in S1 File. We aimed to estimate the factors associated with the exclusion from acknowledgements. In doing so, we reduced the sample to those individuals excluded from authorship, which might introduce a potential risk of sample selection bias. To mitigate this potential bias, we use a Heckman selection model [50]. In the first stage (the selection equation), we use a probit model to estimate selection into Omission from authorship and calculate the inverse Mills ratio. In the second stage (the outcome equation), we also use a probit model and add the inverse Mills ratio to correct for potential bias. The analysis is at the 'dyad-level' (focal scientist-review), so we have multiple observations for each review, depending on the number of authors. Since the same author could appear many times in our sample (e.g., because they contributed to more than one review), we cluster the errors at the author and review level.

To identify the selection equation, we need exclusion restrictions, i.e., variables that explain the binary selection variable (Omission from authorship) but not the dependent variable of the second stage (Omission from acknowledgements). As exclusion restrictions, we use both the difference between the year of publication of the updated review and the original review (*Years difference*) and a dummy variable capturing whether an individual had published in the three years before the publication of the updated review (*No recent publication*). We expect that if the updated version of the review is published after an extended period, there will be more authorship changes; therefore, omission from authorship should be more likely. However, this should not affect the decision to omit an individual from the acknowledgements. In our context, non-authors are acknowledged for a contribution which should be salient to the updated review team, irrespective of the time intercurrent (i.e., they do not need to remember who did what; they can consult the authorship list of the original review). Moreover, some individuals may be missing from the updated review authorship list simply because they are no longer actively publishing, for example, due to parental leave or having left academia. This should not affect whether they are acknowledged or not. To capture this, we checked whether an individual had any publications in the three years before

the publication of the updated review. Given the volumes and timing of publications in this field, not publishing for three years would be atypical.

### Key variables

Following the prior literature, we consider several individual-level characteristics that could shape the attribution of credit.

**Gender.** The authors' gender is represented by two dummy variables (Male and Female). This information was retrieved using Gender-API (https://gender-api.com), which predicts gender based on the first name. This tool has been used in previous studies to allocate binary gender [e.g., 51]. For the unmatched names, we ran a web search using the author's full name to infer gender from their profile pictures on professional web pages. A significant limitation of this approach is that examining these images is likely to help us infer biological factors (such as sex) rather than gender identity. 3% of the individuals were still unmatched and removed from the sample.

**Race.** We used the prediction package Ethnicolr to extract race. This approach has been used in recent studies to categorise racial/ethnic categories of scientists [e.g., 52,53]. Ethnicolr includes several machine learning-based race and ethnicity classifiers trained on different data sets [54]. We used a function that was trained on Florida Voter Registration data. The algorithm takes a list of full names as input to predict, for each name, the probabilities of Asian, Hispanic, Black, and White. We accepted the category with the highest probability as the prediction result. We conducted a web search, considering the profile pictures on their web pages, to identify the race for those records for which the probability of correctly identifying the race was below 75%. This threshold was chosen based on the average confidence of the prediction package in the sample. As expected, the algorithm predicts White names more confidently than Asian, Hispanic, and Black. The average probability that the name provided is of a certain race was 85% for names that are classified as White and 76% for names that are classified as Asian, Hispanic, and Black. We group Asian, Hispanic, and Black scientists into one category called non-White.

**Academic influence.** Scientists' hierarchical position and prior scientific accomplishments positively affect authorship [17]. While acknowledgements have received less attention, previous work suggests similar mechanisms will be at play. For example, Paul-Hus et al. [43] show that acknowledgees are likely to be senior researchers. To measure academic influence, we counted the total number of citations an individual had received one year before the year of publication of the updated review, as indicated in their Scopus record. We divided this by academic age to consider the yearly citation count and used a logarithmic transformation to account for skewness.

**Position.** Medicine largely conforms to contribution-related ordering [55], where the last author is generally a senior scientist, and the early career scientist who implemented the work goes into the first position. The author's position might be linked to attribution of credit; for example, publication authors are less likely to appear as inventors of a related patent when they are the first and last authors [17]. We controlled for the author's position in the original review using three dummy variables: First, Last, and Middle Author(s).

**Institutional status (focal individual).** The prestige of the institution the individual is affiliated with might also shape credit attribution. We examined the affiliation of the focal individual and considered the Times Higher Education (THE) World University Rankings, which identify the strongest universities in different subject areas. We considered the ranking of institutions offering medicine as a subject in 2014/2015, the average year of publication of the updated reviews in our sample. For ease of interpretation, we performed a linear transformation on the rank (400-x), so that the top-ranked institutions are classified as having higher institutional status. For example, the University of Oxford, which was ranked third, we set a value of 397. For the University of Toronto, which was ranked twentieth, we set a value of 380. We assigned a value of 0 if the institution was not in the ranking. For the rare cases with multiple identifiable institutions, following standard academic convention, we retained the first listed university as the primary affiliation.

We also included additional variables to control for the characteristics of the reviews.

**Review stage (original review).** We controlled for whether the original review was at a protocol stage. Research protocol takes the value of 1 if it was only a protocol and 0 otherwise. This distinction is important given that a protocol is simply a plan for the systematic review. At this stage, including an author might be symbolic, as resources have not been invested yet.

**Institutional status team lead (updated review).** Authors at more prestigious institutions might have different attitudes and preferences regarding the attribution of credit. Accordingly, we examined the corresponding author's affiliation and considered the 2014/2015 Times Higher Education World University Rankings.

**External funding (updated review).** Review authors need to acknowledge grants or other forms of support. Sources of support can be internal (provided by the institutions where the review was produced) or external (provided by different institutions, such as funding agencies). We account for this by adding a dummy variable that captures whether the updated review declared receipt of external funding. Receiving external funding might be considered a proxy for quality, as externally funded reviews are more resource-intensive and take longer to complete [56].

**Length of acknowledgements (updated review).** We considered the length of the acknowledgement texts, as defined by the number of characters contained within them and used a logarithmic transformation to account for skewness. Since the length of the acknowledgement texts might be mechanically negatively correlated with our outcome (Omission from acknowledgements), we removed all names from the acknowledgement texts before calculating their length.

**Authors count (updated review).** We included the number of authors of the original review to estimate omission net of the team's size.

**Asia, South America, and Africa (updated review).** Following the approach by Liu et al. [57], we examined the corresponding author's affiliation and considered whether they were located in Asia, South America, and Africa (where most of the population is non-White) or North America, Europe, and Oceania, by including continent dummy variables.

**Difference in race.** Following Tsui et al. [58], we calculated the difference in race as the square root of the summed squared differences between the value on the relevant variable $S$ which stands for the race of the focal individual ($i$) - $S_i$ - and the value on the same variable for every other individual ($j$) in the original review team $S_j$, divided by the count of authors in the team $n$:

$$\text{Difference in race} = \left[ \frac{1}{n} \sum_{j=1}^{n} (S_i - S_j)^2 \right]^{1/2}$$

The resulting score ranges from 0 to approaching (but never reaching) 1. A large score indicates that the focal author differs substantially from other team members. For example, in a team with three non-White authors and one White author, the race difference for the White scientist would be $[(1/4) * [(0–1)^2 + (0–1)^2 + (0–1)^2]]^{1/2} = .43$, but only $[(1/4) * [(0–1)^2]]^{1/2} = .25$ for the non-White scientists.

## Results

We report descriptive statistics in **Table 1**. The sample is well-balanced in terms of gender. As inferred from first names using Gender-API, 45% of the authors of the included reviews were female. However, Asian, Hispanic, and Black scientists, as inferred from their full names using Ethnicolr, represent only 17%, 4%, and 2% of the sample, respectively. This imbalance reflects an uneven pool of Cochrane authors; however, it may also have been amplified by our sampling method. Recall that from the initial sample of 8,267 non-unique authors, we removed 515 authors for whom some information was unavailable (e.g., could not be retrieved in Scopus). Two-sample tests of proportions show that 'Non-White' authors are more prevalent among the excluded authors (34% vs 23%). However, two-sample proportion tests show

**Table 1. Descriptive statistics.**

| Variable | Mean | Std. Dev. | Min | Max |
|---|---|---|---|---|
| Omission from authorship | 0.22 | 0.42 | 0.00 | 1.00 |
| Omission from acknowledgements | 0.08 | 0.27 | 0.00 | 1.00 |
| Female | 0.45 | 0.50 | 0.00 | 1.00 |
| Male | 0.55 | 0.50 | 0.00 | 1.00 |
| Asian | 0.17 | 0.37 | 0.00 | 1.00 |
| Hispanic | 0.04 | 0.21 | 0.00 | 1.00 |
| Black | 0.02 | 0.13 | 0.00 | 1.00 |
| White | 0.77 | 0.42 | 0.00 | 1.00 |
| Academic influence [a] | 42.37 | 87.96 | 0.00 | 1,943.21 |
| First author | 0.26 | 0.44 | 0.00 | 1.00 |
| Last author | 0.26 | 0.44 | 0.00 | 1.00 |
| Middle author | 0.49 | 0.50 | 0.00 | 1.00 |
| Institutional status | 86.40 | 136.04 | 0.00 | 398.00 |
| Research protocol [OR] | 0.04 | 0.20 | 0.00 | 1.00 |
| Year of publication [UR] | 2,014.30 | 3.22 | 2003.00 | 2,019.00 |
| Institutional status team lead [UR] | 106.85 | 145.47 | 0.00 | 397.00 |
| External funding [UR] | 0.65 | 0.48 | 0.00 | 1.00 |
| Length acknowledgements [UR a] | 475.61 | 331.1 | 1.00 | 1,740.00 |
| Authors count [UR] | 4.85 | 2.16 | 1.00 | 18.00 |
| Asia, South America and Africa [UR] | 0.18 | 0.39 | 0.00 | 1.00 |
| North America, Europe and Oceania [UR] | 0.82 | 0.39 | 0.00 | 1.00 |
| Year difference | 5.17 | 3.13 | 0.00 | 21.00 |
| No recent publication | 0.13 | 0.34 | 0.00 | 1.00 |

The sample contains n = 7,752 author-review dyads. (a) raw values are reported; however, these variables were log-transformed prior to analysis. (OR) these variables were calculated for the original review. (UR) these variables were calculated for the updated review.

that 'targeted' omissions were equally frequent in both groups, indicating that the imbalance does not bias our estimates. Moreover, as we discuss later, a statistically significant effect despite the imbalance would strengthen our confidence that the observed effect is not spurious.

## Reviews where at least one contributor was not acknowledged

The sample we use for the estimation contains 2,091 Cochrane reviews. 1,135 reviews had no changes in the authorship team from the original to the updated version, so no previous contributor needed to be acknowledged. Out of the remaining 956 eligible reviews, in which one or more authors left the team in the updated review, 381 (40%) had at least one of the non-author contributors missing from the acknowledgements. The remaining 575 reviews appropriately credited all contributors (**Fig 1**). Reviews omitting some contributors were not all concentrated in certain editorial groups (**Fig 2**). In addition, the proportion of reviews omitting at least one contributor has remained stable over time, with only a marginal decrease after 2014, when we estimate the acknowledgement guidance was introduced (**Fig 3**).

We ran two-sample *t*-tests to compare the means of key variables for reviews where all contributors were acknowledged and those where at least one contributor was not recognised (**Table 2**). The reviews crediting all contributors had longer acknowledgements and higher institutional status, calculated based on the rank of the corresponding author's institution. They were also slightly more recent and more likely to have received external funding. Instead, the reviews that did

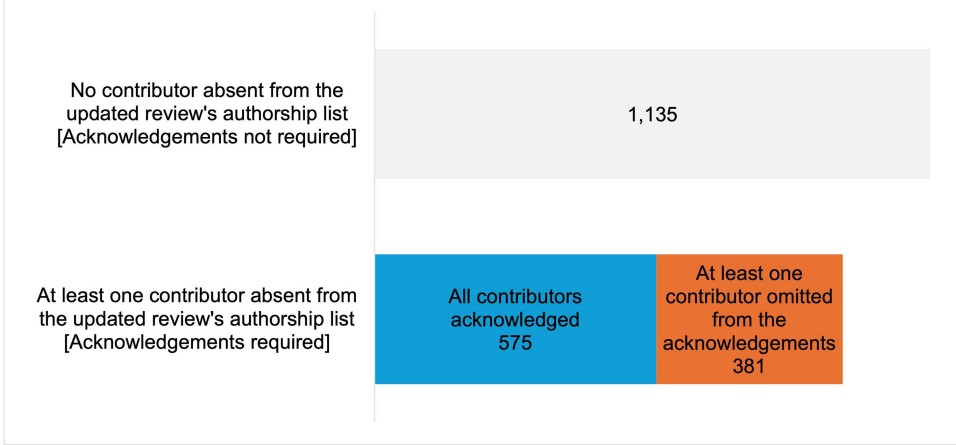

**Fig 1. Percentage of reviews with at least one contributor missing from the acknowledgements – overall.** We show that 381 reviews (40% of those that should have acknowledged prior authors) had at least one contributor missing from the acknowledgements.

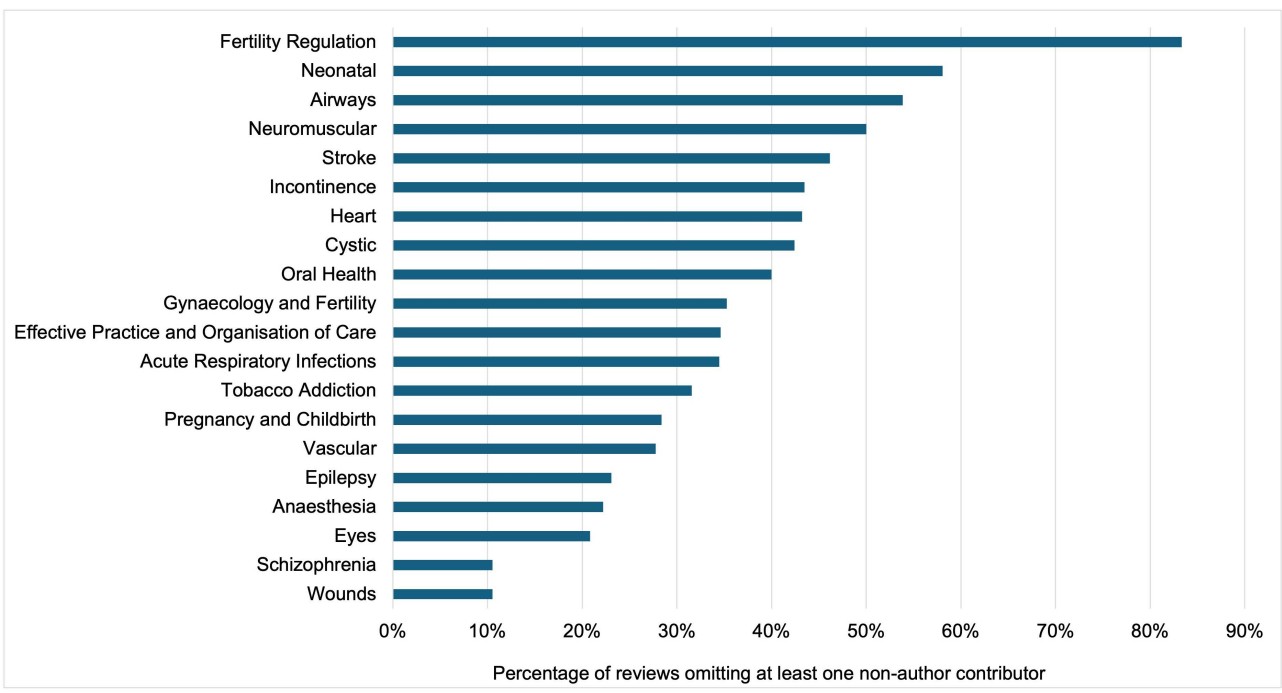

**Fig 2. Percentage of reviews with at least one contributor missing from the acknowledgements – by Cochrane group.** We report the percentage of reviews omitting at least one non-author contributor for the top 20 Cochrane editorial groups based on the number of reviews included in the sample. The Fertility Regulation group had the highest percentage of reviews with at least one contributor not acknowledged (out of those that should have acknowledged prior authors).

not properly acknowledge had larger teams and were more likely to update a review protocol than a full review. Review protocols are simply plans for the systematic review, which describe the review's objectives and methods. Including an author's name at this stage might be symbolic, as resources have not yet been invested; this might explain why protocol contributors are less likely to be acknowledged. Finally, reviews with incomplete acknowledgements were more likely to

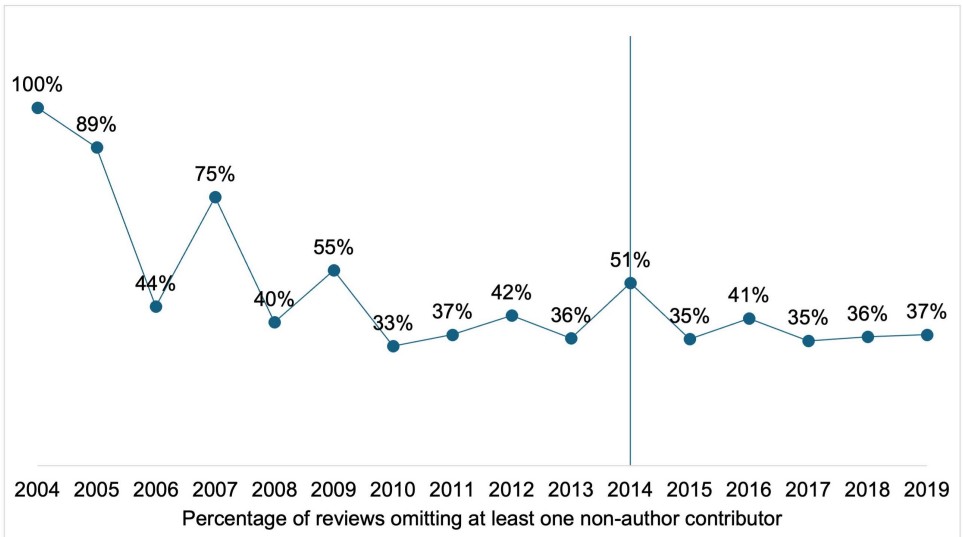

**Fig 3. Percentage of reviews with at least one contributor missing from the acknowledgements – by publication year.** We report the percentage of reviews omitting at least one non-author contributor by year of publication. 41% of the eligible reviews (those that should have acknowledged prior authors) omitted at least one contributor from 2004 to 2013 (before we estimate the guidance was introduced) and 39% from 2014 to 2019 (after we estimate the guidance was introduced).

**Table 2. Reviews where all contributors were acknowledged vs those where at least one contributor was not acknowledged.**

| | All contributors acknowledged (n = 575) | At least one contributor not acknowledged (n = 381) | Difference |
|---|---|---|---|
| Research protocol $^{OR}$ | 0.01 | 0.06 | 0.05*** |
| Institutional status team lead $^{UR}$ | 122.90 | 99.86 | −23.05** |
| External funding $^{UR}$ | 0.67 | 0.61 | −0.06** |
| Length acknowledgements $^{UR}$ | 585.84 | 401.43 | −184.41*** |
| Authors count $^{UR}$ | 4.28 | 5.03 | 0.75*** |
| Asia, South America and Africa | 0.15 | 0.19 | 0.04* |
| Year of publication | 2014.73 | 2014.13 | −0.60*** |

We use two-sample *t*-tests to compare the means of key variables for reviews where all contributors were acknowledged vs. those where at least one contributor was not acknowledged. (OR) these variables were calculated for the original review. (UR) these variables were calculated for the updated review. *** $p < 0.01$, ** $p < 0.05$, * $p < 0.1$.

be from Asia, South America and Africa (as identified by the institutional affiliation of their corresponding author), indicating the presence of geographical differences.

## Contributors omitted from the acknowledgements

At the contributor level, the sample contains 7,752 observations (review-author dyads) corresponding to review authors listed in the original reviews. Of these, 1,732 (22%) were not on the authorship list of the updated review, and 621 (8%) were left off in the acknowledgements of the updated review. These figures indicate that as many as 36% of the authors (621 out of 1,732) missed a mention in the acknowledgements, even though they contributed as authors to the original review but

not to the updated one. We then explore these individuals' characteristics and salient correlates of omission from acknowledgements. We are specifically interested in potential disparities in the acknowledgement of contributions from women and minority groups, for whom a lack of attribution might exacerbate existing inequalities. In **Table 3**, we report odds ratios (ORs), where an OR greater than 1 implies a greater likelihood of being omitted from the acknowledgements.

**Table 3. Correlates of omission from the acknowledgements.**

| Variables | Omission from authorship | Omission from acknowledgements |
|---|---|---|
| Female | 0.97 | 1.01 |
| | (0.04) | (0.05) |
| Non-White | 0.97 | 1.17*** |
| | (0.05) | (0.07) |
| Academic influence | 0.95*** | 1.01 |
| | (0.01) | (0.02) |
| First author | 0.44*** | 0.93 |
| | (0.02) | (0.08) |
| Last author | 0.78*** | 0.94 |
| | (0.03) | (0.05) |
| Institutional status | 1.00 | 1.00 |
| | (0.00) | (0.00) |
| Research protocol OR | 0.73** | 1.13 |
| | (0.10) | (0.21) |
| Institutional status team lead UR | 1.00*** | 1.00** |
| | (0.00) | (0.00) |
| External funding UR | 1.09** | 1.05 |
| | (0.05) | (0.06) |
| Length acknowledgements UR | 1.16*** | 0.74*** |
| | (0.03) | (0.02) |
| Authors count UR | 0.93*** | 1.07*** |
| | (0.01) | (0.01) |
| Asia, South America and Africa UR | 1.07 | 1.01 |
| | (0.06) | (0.07) |
| Years difference | 1.09*** | |
| | (0.01) | |
| No recent publication | 2.18*** | |
| | (0.11) | |
| Inverse Mills Ratio | | 0.36*** |
| | | (0.03) |
| Constant | 0.19*** | 4.09*** |
| | (0.06) | (1.44) |
| Log pseudo-likelihood | −3435.44 | −1810.96 |
| Pseudo-R$^2$ | 0.1657 | 0.1628 |
| Observations | 7,752 | 7,752 |

We show that Non-White contributors were significantly more likely to be omitted than their White counterparts (OR 1.17). Gender and status do not appear to affect omission from acknowledgements. Std. Err. adjusted for clusters by author ID and review ID. Cochrane group dummies and year dummies included. (OR) these variables were calculated for the original review. (UR) these variables were calculated for the updated review. Odds ratio reported. *** p<0.01, ** p<0.05, * p<0.1.

The results for Stage 1 of the Heckman model (Omission from authorship) are generally in line with our expectations, e.g., the first and last authors are less likely to be missing from the authorship list of the updated review than the middle authors. Although we call this Omission from authorship to simplify notation, this might or might not represent an omission, as authors might simply decide to leave the team when a review is updated. Note also that our empirical strategy assumes that someone listed in the original review contributed to it. Linking to issues around 'guest' or 'gift' authorships [59], it is important to note that exceptions might occur, e.g., a named author might not have substantially contributed to the original review.

The results for Stage 2 (Omission from acknowledgements) show that non-White authors were significantly more likely to be omitted from the acknowledgements than their White counterparts (OR 1.17, 95% CI 1.04–1.32, indicating that the odds of omission from the acknowledgements are 1.17 times greater, all the other variables being constant) while Gender does not appear to have an effect. Such omissions were more likely the larger the teams (OR 1.07, 95% CI 1.04–1.10) but less likely the longer the acknowledgements (OR 0.74, 95% CI 0.70–0.78).

Regarding the control variables, a notable finding is that the coefficients for the variables First Author and Last Author are not statistically significant at conventional levels in predicting Omission from acknowledgements. It might be that power dynamics and hierarchical norms that typically shape authorship decisions may be less salient in the case of acknowledgements. There may also be less incentive to include senior individuals out of deference or obligation, as acknowledgements are less closely tied to career outcomes or prestige than authorship. Moreover, the rules of attribution in the case of the acknowledgements are looser relative to authorship, which might give more autonomy to the author and reduce the influence of seniority or institutional politics.

Following Certo et al. [60], we examine the correlation between our key regressors and the Inverse Mills Ratio as an indicator of the strength of the exclusion restrictions. The low correlations (ranging from 0.031 to 0.047) suggest that the chosen exclusion restrictions are effective. At the same time, the increase in the pseudo-$R^2$ of the first-stage model when including the exclusion restrictions (from 0.11 to 0.16) indicates that the exclusion restrictions enhance the model fit.

### Further analysis: Non-White contributors

We investigated further the potential factors shaping the disadvantages of non-White contributors (see Table A in S2 File). First, we considered potential intersectional inequalities [61]. We found that non-White male contributors were more likely to be omitted than their White male counterparts (OR 1.24, 95% CI 1.07–1.44). However, note that there are only a few observations in the non-White female contributors' subgroup (n = 676, corresponding to 8.7% of the sample).

Second, certain types of collaborators might be disproportionately associated with technical tasks instead of conceptual ones [62,63], which might impact their odds of being acknowledged. Thus, we considered the contribution made by the focal individual to the original review. We split each contribution statement into shorter strings, each including an author-contribution tuple. We separated the names from these strings and matched them against the authorship list. We then extracted keywords that could indicate "minor/technical" and "major/conceptual" contributions (Table B in S2 File). A string containing only minor contributions keywords was deemed a "Minor/technical contribution". For example, take a review with authors AJ, CH and EF and the following contribution statement: "AJ contributed to the development of the protocol. CH performed the data extraction. EF assessed identified studies for eligibility". AJ and EF would be categorised as having made a major contribution, and CH as having made a minor one. Using these methods, we could attach a minor or a major contribution to 6,793 authors (88% of the sample). We tried to address the third case by running manual checks and adjustments. Table C in S2 File reports a sample of acknowledgement texts. Non-White contributors in our sample were more likely to have made a minor/technical contribution (two-sample tests of proportion, 0.13 versus 0.11, p < 0.10). However, when we examined the combined effects on predicting omission from the acknowledgements, we found that relative to White contributors who made a major contribution, only non-White contributors who made a major

 

contribution were more likely to be omitted. Note that this is true only at the 10% significance level, as the 95% CI includes 1 (OR 1.15, 95% CI 1.00–1.31).

In further analysis, reported in Table D and Table E in S2 File, we also examined (1) race differences within the team [64], (2) homophily between the focal scientist and the team's lead, (3) the location of the corresponding author of the updated review to assign the focal review to North America, Europe and Oceania or Asia, South America, and Africa, and finally (4) the location of the focal author to assign their location to North America, Europe and Oceania or Asia, South America, and Africa. We find that our key result (non-White contributors were more likely to be omitted from the acknowledgements) cannot be explained by these contributors being in teams that are predominantly White or led by White scientists. The result appears to be shaped by geographical disparities, with non-White contributors being more likely to be deprived of due acknowledgements in reviews from Asia, South America and Africa.

### Distribution of contributors omitted from the acknowledgements

In each review, omission from the acknowledgements might affect only some or all contributors (see Table C in S2 File). In line with norms of distributive justice [65], the former case is more likely to be perceived as unfair by the omitted contributors because comparisons with their collaborators would reveal that they are relatively under-rewarded. We explored this descriptively. Out of the 621 contributors excluded from the acknowledgements, only 72 (12%) were found in reviews where only some deserving contributors were omitted. The remaining 549 (88%) were found in reviews where all deserving contributors were omitted. These figures indicate that targeted exclusions are rare. Instead, most exclusions appear to be concentrated in a review with poor acknowledgements.

We also compared the distribution of the key characteristics of interest (Female, Non-White) across these two groups. There was no significant difference in the proportion of female contributors; however, non-White contributors were more represented in the "wholesale omission" category—cases where no prior contributors were acknowledged at all. This pattern confirms our intuition that the observed disparities are driven more by weak or inconsistent acknowledgement practices than by deliberate, selective exclusion of specific groups. The results of these tests are reported in Table F in S2 File.

### Robustness checks

We ran several robustness checks (Table G in S2 File). First, the distribution of the yearly citations for the authors included in our sample is highly skewed. To check whether our results are robust to removing highly cited individuals from the sample, we removed individuals in the top 5% of the yearly citations distribution. Second, review protocols might be informally subject to different norms. For example, authorship might be more loosely defined when a project has not started yet. Thus, we removed review protocols from the sample to investigate whether this influences the relationships we explore. Third, the gender recognition software we use is more accurate with specific names. A comparative study by Sebo [66] found Gender-API to be one of the most accurate tools, with a low proportion of misclassifications (0.8% misclassified male and 2.2% misclassified female) and a low proportion of unrecognised names (0.3%). Yet, as discussed above, the software is not entirely accurate, especially for names using non-Latin alphabets. To test the robustness of our findings, we removed from the sample those records where the probability that the gender was correctly identified was lower than 50%, as determined by Gender-API.

Fourth, to capture academic productivity beyond academic influence, we added controls for "Productivity" (as measured by the yearly publications of the focal individual) and "Cochrane experience" (cumulative Cochrane publications of the focal individual). Fifth, we checked whether the inclusion of a new variable, "Acknowledges others besides prior review authors", which might be seen as a measure of the generosity of the updated review team, would affect your current results. These checks confirm that our key finding remains robust even with the inclusion of these controls.

Moreover, our main analysis considers non-White as one category due to the limited number of contributors in each subcategory. To try to parse the individual effects of each subgroup, we entered them in the model separately. When we entered Asian, Hispanic, and Black as separate variables into the model, we found that only Asian and Hispanic authors were more likely to be omitted than White authors. We note, however, that these subgroups have few observations. Moreover, the Hispanic category is particularly problematic as the prediction package is trained on US data, and our sample includes an international pool of scientists.

Finally, to correct for potential underrepresentation of the non-White group in our sample, in further checks, we employed the pweight = weights specification in STATA to assign probability weights to the non-White group. Our key finding was robust to the inclusion of these weights. We also note that this imbalance makes it more difficult to detect a significant association between race and omission from the acknowledgements. Since our sample includes a much larger proportion of White authors than non-White authors, the variance of our key independent variable (i.e., the 'Non-White' dummy) is limited, thereby reducing statistical power. Therefore, a statistically significant effect despite the imbalance strengthens our confidence that the effect we observe is real rather than spurious.

## Discussion

Using data from 2,091 Cochrane reviews, we observed the omission of non-author contributors from the acknowledgements of updated reviews. This setting helps us to overcome prior research's limitations in unpacking attribution and contribution and allows us to determine which non-author contributors were denied credit despite deserving an acknowledgement. We found that 40% of the eligible reviews in our sample (i.e. those that should have acknowledged prior authors) did not appropriately acknowledge non-author contributors. This proportion remained relatively stable even after the introduction of the acknowledgement guidance. We explored whether omission from the acknowledgements correlated with salient individual-level characteristics of the collaborators, with a specific focus on gender and race. Female and male contributors had similar odds of omission. However, non-White contributors were more likely to be omitted from the acknowledgement relative to their White colleagues. This is concerning as the lack of attribution for these contributors might amplify existing inequities in opportunities and visibility [e.g., 67].

The positive association between race and omission from the acknowledgements cannot be fully explained by intersectionality, the type of review task, being a race minority in the team, and homophily effects. Non-White contributors were less likely to be missing from the acknowledgements in White-dominated teams but more likely to be 'forgotten' when the corresponding author was non-White. The team lead's location also plays a role. Non-White authors in reviews from Asia, South America, and Africa were more likely to be omitted from the acknowledgements than White authors in reviews from North America, Europe, and Oceania. This suggests that geographical disparities may contribute to the attribution disadvantages faced by non-author contributors from minority groups. Importantly, in most cases, under-recognition affected all review contributors, rather than just some. This indicates that the disparities we observe likely reflect poor acknowledgement practice rather than the targeted exclusion of certain groups of contributors.

### Implications for research

Our findings have implications for research on the functioning of the attribution of credit system and inequalities in science. First, we shed light on the recognition of non-author contributors in the acknowledgements. While disparities in authorship and citations have been extensively reviewed, acknowledgements have received comparatively little attention. Given the increasing importance of non-author collaborators, ensuring that every research contribution is appropriately acknowledged is essential to promote transparency and fairness [e.g., 68]. Moreover, the indexation of acknowledgement texts has increased interest in this topic, and the study of acknowledgements is seen as a growing field [14,24,69] with several applications, including network analysis [e.g., 70–72].

Second, we contribute to research on inequalities in science by providing evidence that the lack of recognition of minority groups, particularly non-White scientists, is also manifest through the omission of their names in acknowledgements. Overcoming prior work's limitations in unpacking attribution and contribution, our setting allows us to determine which non-author contributors were denied credit despite having made a contribution. Our results confirm that leveraging the "*stratified nature of scientific documents*" ( [30]: p. 36) and using information from different types of para-text, including acknowledgements, can help generate richer insights into inequalities in science. Overall, we document only weak links between gender and attribution of credit in this setting, which we interpret as a positive outcome.

Our findings suggest that the link between social factors, including mechanisms related to gender and race, and the attribution of credit in acknowledgements might be weaker than for authorships. While adding extra names to the authorship list might raise concerns about the dilution of contributions, research teams might have little to lose by being generous in their acknowledgements.

The lack of strong effects might also relate to the type of task being acknowledged in our context. Prior research has shown that when there is task ambiguity, such as when there is no tangible outcome, evaluations of minority groups' competencies are more likely to be negatively affected. For example, Heilman and Haynes [73] found that female members in mixed-gender teams are devalued unless ambiguity about their contributions is reduced. Similarly, Sarsons [74] showed that women receive less credit for work when their assistance is unclear (i.e., when they co-author as opposed to publishing on their own). The task being acknowledged in the case of updated Cochrane reviews is relatively well-defined, i.e., one has either contributed to the previous review or has not. As such, potentially negative expectations about minority groups' contributions might be mitigated by clear evidence of prior work.

Third, our study offers insights into the dynamics of attribution of credit that occur when a research project endures over time and relies on the efforts of contributors who may have left the team. Much of the extant work on the factors associated with attribution of credit has been concerned with publication, or patents, at one point in time [17,75]. Yet, as science is moving towards 'living' research projects [49] and increasing involvement of end users and crowd science approaches [76], it is important to consider how the norms of attribution apply to more dispersed projects with an open-ended date. For example, fairness expectations with regard to both the distribution of value ('distributive fairness') and the procedures leading to such distribution ('procedural fairness') can affect potential contributors' decision to participate in crowdsourcing [77]. The issue of who is 'forgotten' when a project evolves is particularly significant for minority group contributors, who have been shown to have less sustainable careers in academia, e.g., with gender differences being explained by career lengths and dropout rates [19]. Furthermore, these omissions might carry over to other types of rewards, as acknowledgements are related to citations and co-authorship [36].

## Implications for practice

This study contributes to a deeper understanding of editorial guidance and practices related to contributorship and credit attribution. In the context of Cochrane reviews, incomplete attribution in the acknowledgement is found despite the guidance. Such disparities could be reduced if all authors were made aware of and followed the guidance. Our results, particularly those related to geographical disparities, underscore the importance of increasing the visibility of new guidelines to ensure adherence and the necessity of formulating targeted training plans. It is somewhat surprising that some reviews do not comply with the guidance, given that the Cochrane network is highly active in organising meetings and workshops. We would expect much lower compliance in other contexts, even within the medical field; hence, our results are likely to represent lower-bound estimates of what one would observe in those settings.

Cochrane or any groups putting forward contributorship recommendations (e.g., the ICMJE) should ensure widespread dissemination of information concerning proper attribution of non-author contributors, including translation into different languages. Once awareness is widespread, adherence could be strengthened, for example, by making the guidance

mandatory. Failure to acknowledge deserving contributors is a poor research practice; therefore, more training on ethical and fair attribution would also be beneficial.

Our results suggest several pragmatic steps to reduce omissions in acknowledgements. First, editorial teams could incorporate a brief 'prior contributors' checklist into author guidelines for updated reviews, prompting teams to verify that all prior authors are either retained as authors or acknowledged as contributors. Second, journals and review groups could offer short micro-guidance and training on acknowledgement norms, especially for multi-national teams where conventions vary. Third, an optional structured acknowledgement field (e.g., 'prior-review contributors') would simplify compliance and make it auditable. These measures are low-cost, compatible with existing workflows, and aligned with ICMJE/ Cochrane guidance, thereby improving transparency and fairness without imposing an undue burden.

The observed disparities in acknowledgement credit point to the need for more transparent, accountable authorship and acknowledgement procedures within research organisations and journals. Establishing structured contributor-role statements, routine monitoring of acknowledgement patterns, and explicit inclusion guidance could reduce inadvertent omissions and improve recognition for underrepresented groups. Publishers and funders could integrate these practices into their reporting requirements, while research institutions could embed acknowledgement norms within their responsible research training and equity policies. Together, such measures would strengthen transparency and fairness in how credit is distributed across collaborative research teams.

Attribution of credit in the acknowledgements is generally defined more loosely than in authorship (e.g., a mention in the acknowledgements is a voluntary act, rarely enforced by journals or publishers). In the case of Cochrane, the opposite appears to be true because acknowledgement rules are clearly defined. Journals outside the medical field where contributorship practices remain largely implicit should consider adopting similar guidance. This is becoming increasingly important as the creation of acknowledgement databases is likely to drive attention to new metrics, such as the count of acknowledgement mentions, which may be used to measure a researcher's intellectual contribution.

The study also has implications for authors and non-author contributors. The expectation that non-author contributors have different norms and interests should be challenged [e.g., 25,32]. A mention in the acknowledgements might not necessarily be associated with the accumulation of social capital, yet misattribution of credit in acknowledgements can undermine the allocation of important resources [e.g., 33].

Finally, acknowledgement research is gaining traction as a developing area of study. Researchers who plan to use acknowledgement texts to identify informal idea exchanges should be aware that inequalities may extend to the contributors recognised in the acknowledgements.

## Limitations and future research

Our analysis is limited to the study of updated Cochrane reviews. This setting is unique in at least three dimensions: the subject field (healthcare), the type of studies under investigation (systematic reviews), and the guidance and format for acknowledgements (formal rules). As contributorship practices and the prevalence of non-author collaborators on authorship lists vary not only across fields but also according to study characteristics [78,79], contributorship norms in Cochrane reviews might differ from those of traditional research articles [80]. Regarding acknowledgements' guidance and format, it is important to consider that we examine specific forms of acknowledgements (i.e., recognising prior authorship) as opposed to other contributions, such as administrative support or editing/proofreading. Moreover, while acknowledgement guidance exists beyond the medical field, for example, it is adopted in social science by the Campbell Collaboration, adherence and monitoring efforts are varied. Taken together, the generalisability of our findings to different types of publications, contributors, and fields should, therefore, be explored.

Our data includes reviews that were updated as of June 2019. Cochrane has maintained their policy of regularly updating their reviews, and at the time of writing, we are not aware of any new Cochrane guidance or training regarding acknowledgements. Moreover, as we estimate that the guidance has been well-known within the Cochrane network since

2014, the time period we consider should be sufficiently long to capture any delays in the uptake of the guidance. Nevertheless, we cannot exclude the possibility that the authorship and acknowledgement structures we attempt to capture would differ in the absence of this temporal cut-off.

Moreover, we cannot unpack why specific non-author contributors were omitted from the acknowledgements. For example, minority group scientists might be less likely to self-promote or have weaker ties with the author teams. Qualitative research approaches, such as in-depth interviews and surveys, might help explore these dynamics and motivations in more depth. Our study leverages gender and race, relying on prediction packages that can be highly inaccurate and fail to address diverse gender identities and modalities. Richer — for example, non-binary — measures might be considered in future work, enriching the data by asking for self-identification of gender and race. Future research could also examine other factors related to unwarranted exclusion, such as homophily in discipline, affiliation, academic age, or geographic separation of collaborators [81,82]. As only a few female scientists are in the non-White category (8% of the sample), race and gender effects might be difficult to disentangle for this group. Larger datasets might allow us to better unpack effects at the intersection of gender and race. Finally, future work could improve our analysis using automatic acknowledgements sections classification performed with machine learning [83].

Despite these limitations, we hope this study will provide new insights into the functioning of the attribution of credit system and, more specifically, the acknowledgement of contributors beyond the authorship team.

## Supporting information

**S1 File. Supplemental figures.**
(DOCX)

**S2 File. Supplemental tables.**
(DOCX)

## Acknowledgments

We thank Stefano Benigni, Christina Cohrs, Chandresh Gupta, Damiano Morando and Virendra Velagapudi for assistance with data collection and coding. The paper also benefited from feedback from participants at: Workshop on Medical Innovation and Healthcare (WOMI) 2020, University of Bath Centre for Research on Entrepreneurship and Innovation (CREI) Paper Development Workshop series 2022, Bologna Business School PhD Seminar series 2022, Cochrane Colloquium 2023, DRUID 2023, Politecnico di Milano Seminar series 2023 and Rotterdam School of Management Seminar series 2024.

## Author contributions

**Conceptualization:** Rossella Salandra, Marisa Miraldo, Paola Criscuolo.

**Data curation:** Rossella Salandra.

**Formal analysis:** Rossella Salandra.

**Methodology:** Rossella Salandra, Marisa Miraldo, Paola Criscuolo.

**Project administration:** Rossella Salandra.

**Writing – original draft:** Rossella Salandra, Marisa Miraldo, Paola Criscuolo.

**Writing – review & editing:** Rossella Salandra, Marisa Miraldo, Paola Criscuolo.

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
