## [Decision Letter · Decision Letter 0]

22 Sep 2025

Dear Dr. Salandra,

Your manuscript was sent to three reviewers with strong and complementary expertise. I also read the manuscript independently and I agree with their assessment. This is a well-conceived and well-written paper, however, there is room to strengthen the manuscript on several fronts.

On the empirical side, each reviewer highlights very actionable aspects to make the results more robust. Please address these points carefully. Regarding the conceptual framework, the paper would benefit from a more developed background discussion on the role and use of acknowledgements in scholarly communication. In its current form, the “Related Work” section touches on central aspects but does not fully elaborate them. Expanding this section by engaging with themes such as the implicit codes of professional and normative behaviour acknowledgements follow, the variation in co- and sub-authorship practices across research domains, and the under-recognised role of acknowledgements within the reward system of science, would significantly strengthen the positioning and contribution of the paper. I provide some references below that may be helpful in this regard.

Baccini, A., & Petrovich, E. (2021). Normative versus strategic accounts of acknowledgment data: The case of the top-five journals of economics. Scientometrics. https://doi.org/10.1007/s11192-021-04185-6

Cronin, B. (1995), The Scholar’s Courtesy: The Role of Acknowledgments in the Primary Communication Process, Taylor Graham, Los Angeles, CA.

Díaz-Faes, A. A., & Bordons, M. (2017). Making visible the invisible through the analysis of acknowledgements in the humanities. Aslib Journal of Information Management, 69(5), 576–590. https://doi.org/10.1108/AJIM-01-2017-0008

Hyland, K. (2003), “Dissertation acknowledgments: the anatomy of a Cinderella genre”, Written Communication, Vol. 20 No. 3, pp. 242-268.

Mccain, K. W. (1991). Communication, Competition, The Production and Secrecy : and Dissemination of Research-Related in Genetics Information. Science, Technology, & Human Values, 16(4), 491–516.

We look forward to receiving your revised manuscript.

Kind regards,

Adrian A. Diaz-Faes, PhD

Academic Editor

PLOS ONE

Journal Requirements:

5. We note that there is identifying data in the Supporting Information file <S1 Figure.docx>. Due to the inclusion of these potentially identifying data, we have removed this file from your file inventory. Prior to sharing human research participant data, authors should consult with an ethics committee to ensure data are shared in accordance with participant consent and all applicable local laws.

-Location data

6. If the academic editor or reviewer comments include a recommendation to cite specific previously published works, please review and evaluate these publications to determine whether they are relevant and should be cited. There is no requirement to cite these works unless the editor has indicated otherwise. 

Reviewers' comments:

Reviewer's Responses to Questions

**Comments to the Author**

1. Is the manuscript technically sound, and do the data support the conclusions?

Reviewer #1: Yes

Reviewer #2: Yes

Reviewer #3: Partly

2. Has the statistical analysis been performed appropriately and rigorously?

Reviewer #1: Yes

Reviewer #2: Yes

Reviewer #3: Yes

3. Have the authors made all data underlying the findings in their manuscript fully available?

Reviewer #1: Yes

Reviewer #2: Yes

Reviewer #3: Yes

4. Is the manuscript presented in an intelligible fashion and written in standard English?

Reviewer #1: Yes

Reviewer #2: Yes

Reviewer #3: Yes

Reviewer #1: The paper makes an important contribution to the study of inequalities in science, supplementing prior work on recognition in authorships and citation by gender, race, region, etc. by looking at acknowledgements.

Cochrane reviews provide a good setting for looking for demographic differences in acknowledgment.

You might also want to discuss more why seniority/position does not affect acknowledgement, even though it effects authorship (cf. Lissoni, et al., 2013, which you cite).

Controlling for average cites is good. How about controlling for total pubs (or total in last three years) as another measure of visibility?

How robust are your result to recoding the generic acknowledgements (“We acknowledge the work of the author team for the first version of the review”.) as 0 instead of 1?

What happens if you add a new variable, “Acknowledges others besides prior review authors”? And see what review team variables predict this (same as what predicts dropping prior authors, but with sign reversed)? And, if you then control for this variable, does it affect your current results? This can be seen as a measure of how generous/stingy the author of the new review is, and might explain some of the drops in acknowledgements (beyond the demographics of prior authors).

For the effect of length, you might want to exclude the author names from the length count, as these are mechanically correlated with the outcome (in the wrong direction).

“Following Certo et al. [43], to assess the strength of our exclusion restrictions, we checked) the correlation between the key independent variables of interest in the second stage (Female and Non-White) and the Inverse Mills Ratio, as well as the pseudo-R2 associated with the first stage.“

What do the correlations show?

What happens when you control for the continent of the Original Review? You report the ContinentxRace interactions in the Further Analyses. But, what is the main effect of the original review being from Asia-South America-Africa? Put differently, are the URs discriminating on the basis of race or geography?

At the same time, the heterogeneity analyses about the race/gender X team characteristics is not well motivated and seems to distract from the key findings. If you can drop this, or limit to the Supplement, it might be better.

Is it possible to do an analysis of just the discerning omissions? Yes, the Ns get small, but this might be informative. But, showing wholesale exclusion versus discerning exclusion can address, for example, your conjectures about UR's not knowing norms versus URs discriminating against certain types of authors.

At the same time, it is an unusual setting, and so you need to strongly acknowledge the limitations on generalizability (for example, see Jabbehdari and Walsh, 2017, STHV, for field differences in non-author collaborators).

Reference:

Authorship Norms and Project Structures in Science

S. Jabbehdari and J. P. Walsh

Science, Technology, & Human Values 2017 Vol. 42 Issue 5 Pages 872-900

Reviewer #2: In this study, the authors use a corpus of approximately 2,000 Cochrane systematic reviews in medicine to investigate fairness in the attribution of credit in science, focusing in particular on the relationship between acknowledgments and formal authorship. They examine the factors that lead to omission of credit by analyzing the racial and gender characteristics of omitted contributors, and they show that geographical disparities—rather than explicit biases against specific communities—affect the likelihood of omission.

I particularly appreciate the choice of the Cochrane corpus: the specific authorship and acknowledgment guidelines adopted by Cochrane enable to unpack the relationship between attribution and contribution in a way that is not possible with standard scientific publications. I also commend the substantial manual work involved in matching acknowledgees with authors and resolving ambiguities, which ensures the high quality of the dataset.

I also value the authors’ attempt to measure the contribution of acknowledgees through keyword analysis. Furthermore, the discussion is broad and engaging, situating the study’s findings within the broader evolution of science toward more collective practices (e.g., citizen science) and exploring the challenges of fairly recognizing contributions in these contexts. Finally, the statistical methods used are robust.

Before proceeding to publication, the following minor issues should be addressed:

• Streamline the literature review to reduce overlap with the introduction.

• Clarify how the authors’ profiles were retrieved from Scopus: was this done manually or via the Scopus API?

• Page 9: it is unclear whether the race attribution algorithm was independently validated by manually checking a sample of authors. Is the success rate of below 75% mentioned in the text the result of this validation?

• Specify how the variable Institutional Status was modeled: does it refer to the university’s position in the Times Higher Education ranking? How were cases handled where authors had multiple affiliations?

• Table 1: clarify whether the reported statistics for the variables marked with “a” are log-transformed values or raw values before transformation.

• On page 21, among the studies applying network analysis to acknowledgments data, it would be appropriate to mention [DOI: 10.1007/s11229-022-03515-2].

Reviewer #3: This study explores whether acknowledgements in academic publications omit deserving contributors and examines the heterogeneity of these omissions by gender and race. It raises interesting questions. I would suggest clarifying the following points to enhance the manuscript's clarity.

1. First, the paper needs to improve its formatting for better readability, ensuring that both paragraphs are aligned.

2. The authors need to clarify the significance of their findings. What can be done by discovering the differences in acknowledgements related to race and gender?

Introduction

3. The authors should clarify how they determine the racial and gender identities of contributors, as well as the accuracy of the tool they use, in the introduction.

4. Emphasizing the contributions of this work is crucial; while the authors present observations and data, they should highlight the broader implications of these findings.

5. I suggest to summarizing the research questions at the end of the introduction may also enhance clarity.

Related work

6. The related work section needs to be more detailed and multi-dimensional. I suggest adding subheadings to enhance logical flow and coherence. It would also be beneficial to include a section on "The Impact of Race and Gender on Acknowledgement Practices." Additionally, more references should be incorporated regarding studies that use names to identify gender or race. Also, find related research that using Cochrane dataset.

Method

7. Create a framework figure to illustrate the methodological approach, aiding comprehension for readers.

8. Why was the sample date created using reviews updated as of June 2019?

9. There are 2,091 corresponding to 8,267 non-unique authors, but the same author name may appear in multiple reviews, representing different individuals. How should the issue of authors with the same name be addressed?

10. A flowchart would be beneficial to visually represent the methodological approach and decision-making process. Please include one.

Result

11. How to deal with the problem of data imbalance. Asian is 0.17; White is 0.77.

12. The authors should avoid grouping multiple parts of a figure (e.g., Figure 1 A, B, C) into one and instead map distinct titles to each sub-figure for clarity.

13. The discussion section can also include subheadings.

**Do you want your identity to be public for this peer review?** For information about this choice, including consent withdrawal, please see our Privacy Policy

Reviewer #1: No

Reviewer #2: **Yes: ** Eugenio Petrovich

Reviewer #3: No

---

## [Author Response · Author response to Decision Letter 1]

28 Oct 2025

All responses are included in the "Response to Reviewers" file. Many thanks

---

## [Decision Letter · Decision Letter 1]

27 Nov 2025

Attribution of credit in acknowledgements: The case of systematic reviews in medicine

PONE-D-25-29842R1

Dear Dr. Salandra,

We’re pleased to inform you that your manuscript has been judged scientifically suitable for publication and will be formally accepted for publication once it meets all outstanding technical requirements.

Kind regards,

Adrian A. Diaz-Faes, PhD

Academic Editor

PLOS ONE

Additional Editor Comments (optional):

Thank you for your work on the paper. You have thoroughly addressed all the comments, both in the background and empirical sections. I believe this is a well-rounded and well-positioned manuscript that should be of interest to scholars in the field of the science of science

Reviewers' comments:

Reviewer's Responses to Questions

**Comments to the Author**

Reviewer #1: All comments have been addressed

Reviewer #2: All comments have been addressed

Reviewer #3: All comments have been addressed

2. Is the manuscript technically sound, and do the data support the conclusions?

Reviewer #1: Yes

Reviewer #2: Yes

Reviewer #3: Yes

3. Has the statistical analysis been performed appropriately and rigorously?

Reviewer #1: Yes

Reviewer #2: Yes

Reviewer #3: Yes

4. Have the authors made all data underlying the findings in their manuscript fully available?

Reviewer #1: Yes

Reviewer #2: Yes

Reviewer #3: Yes

5. Is the manuscript presented in an intelligible fashion and written in standard English?

Reviewer #1: Yes

Reviewer #2: Yes

Reviewer #3: Yes

Reviewer #1: The authors engaged in substantial revision and have sufficiently addressed all the comments of the editor and of the reviewers.

Reviewer #2: (No Response)

Reviewer #3: (No Response)

**Do you want your identity to be public for this peer review?** For information about this choice, including consent withdrawal, please see our Privacy Policy

Reviewer #1: No

Reviewer #2: **Yes: ** Eugenio Petrovich

Reviewer #3: No

---

## [Editor Report · Acceptance letter]

PONE-D-25-29842R1

PLOS One

Dear Dr. Salandra,

I'm pleased to inform you that your manuscript has been deemed suitable for publication in PLOS One. Congratulations! Your manuscript is now being handed over to our production team.

Kind regards,

on behalf of

Dr. Adrian A. Diaz-Faes

Academic Editor

PLOS One